# The Economic Impact and Health-Related Quality of Life of Spinal Muscular Atrophy. An Analysis across Europe

**DOI:** 10.3390/ijerph17165640

**Published:** 2020-08-05

**Authors:** Luz María Peña-Longobardo, Isaac Aranda-Reneo, Juan Oliva-Moreno, Svenja Litzkendorf, Isabelle Durand-Zaleski, Eduardo Tizzano, Julio López-Bastida

**Affiliations:** 1Department of Economic Analysis and Finance, University of Castilla-La Mancha, Cobertizo San Pedro Martir S/N, 45071 Toledo, Spain; juan.olivamoreno@uclm.es; 2Department of Economic Analysis and Finance, University of Castilla-La Mancha, 45600 Talavera de la Reina, Spain; Isaac.aranda@uclm.es; 3Centrer for Health Economics Research Hannover, Leibniz University Hannover, 30159 Hannover, Germany; sl@cherh.de; 4Department of Research in Clinic of Health Economics, Université de Paris, CRESS, INSERM, INRA, 75000 Paris, France; isabelle.durand-zaleski@aphp.fr; 5Department of Clinical and Molecular Genetics and Medicine Genetics Group, Vall d’Hebron University Hospital, 08001 Barcelona, Spain; etizzano@vhebron.net; 6Faculty of Health Science, Talavera de la Reina, University Castilla-La Mancha, 45600 Talavera de la Reina, Spain; Julio.LopezBastida@uclm.es

**Keywords:** spinal muscular atrophy, health-related quality of life, economic burden, cost-of-illness, Europe, informal care

## Abstract

*Background:* this study aimed to estimate the economic impact and health-related quality of life (HRQOL) of patients with spinal muscular atrophy (SMA) in three European countries. It was used a cross-sectional study carried out in France, Germany, and the United Kingdom. Data were collected from July 2015 to November 2015. Healthcare costs (hospitalizations, emergencies, medical tests, drugs used, visits to general practitioners (GPs) and specialists, medical material and healthcare transport), and non-healthcare costs (social services and informal care) were identified and valued. EuroQol instruments, the Zarit interview, and the Barthel Index were also used to reflect the burden and the social impact of the disease beyond the cost of healthcare. *Results:* we included 86 children with SMA, 26.7% of them had Type I, and 73.3% Type II or III. The annual average cost associated with SMA reaches €54,295 in the UK, €32,042 in France and €51,983 in Germany. The direct non-healthcare costs ranged between 79–86% of the total cost and the informal care costs were the main component of these costs. Additionally, people suffering from this disease have a very low health-related quality of life, and there are large differences between countries. *Conclusions:* SMA has a high socioeconomic impact in terms of healthcare and social costs. It was also observed that the HRQOL of affected children was extremely reduced. The figures shown in this study may help to design more efficient and equitable policies, with special emphasis on the support provided to the families or on non-healthcare aid.

## 1. Introduction

Studies of the economic impact of a disease, commonly known as cost-of-illness studies, are a type of analysis that is well known and disseminated in the scientific literature in the field of health economics. The interest of such studies lies in revealing an insufficiently known dimension of a disease—its economic burden—and in incorporating this information into the body of knowledge about it. Thus, in the field of health economics, these studies are the equivalent of epidemiological studies in the field of public health. Moreover, although they do not allow us to identify the most effective or efficient interventions in a specific disease, they do help to raise awareness of its social impact [1,2,3,4,5,6,7].

Although, in the field of high-prevalence diseases, the presence of this type of studies is frequent and growing [7,8], this is not the case in the field of rare diseases, due to the inherent difficulty of obtaining information about the people who suffer from them. Even though, in recent years, efforts have been made to find more information about the economic burden posed by rare diseases [9], there is still a serious lack of information about many of them. First, due to their low prevalence, the correct diagnosis of rare diseases is complex and subject to significant delays. Moreover, most rare diseases have no cure, and, for many, either there is no effective treatment available or, if treatments do exist, there is no guarantee of improvement in life expectancy or quality of life.

Several factors could explain why this disease has such a strong social impact on sufferers. They are its severity, uncertainty in the diagnosis, and the lack of effective treatments. They suffer from degenerative and life threatening, not only for the sufferers, but also for their families. It has been shown in the literature that approximately 50% have onset in childhood, and over one-third of deaths of children under one year old are due to rare diseases [9,10]. The health-related quality of life (HRQOL) of people who suffer from these diseases is also seriously threatened. The results of recent studies are coincident in concluding that people suffering from these diseases show results well below those of the general population, as several dimensions of the HRQOL being affected simultaneously [11].

One very common rare disease is spinal muscular atrophy (SMA). SMA is an autosomal recessive neuromuscular disorder caused by the degeneration of alpha motor neurons in the anterior horns of the spinal cord. It is caused by homozygous absence or pathogenic variants in the survival motor neuron gene 1 (SMN1), and the phenotype is mainly influenced by the number of copies of a highly homologous gene, survival motor neuron gene 2 (SMN2), present in all patients. Weakness is the most important manifestation, with several complications such as respiratory insufficiency, scoliosis, contractures, and nutritional problems. SMA is classified in three main types according to age of onset and motor milestones achieved. Type I starts in the first weeks or months of life, the patients never sit, and death occurs in the first two years of life. Type II manifests after 6 months, patients never walk and are wheelchair-bound for life. And Type III appears after 18 months and patients may walk for several years but may lose this ability later [12].

In fact, SMA has an incidence of 1/5000 to 1/10,000 births and a carrier frequency of 1/35 to 1/50 [13], being one of the most severe hereditary diseases among children. Furthermore, the disability caused by this disease increases the difficulty of carrying out the activities of daily living (ADL) and the burden borne by the families [11]. Therefore, in order to quantify the real economic burden of this disease, it is therefore necessary to take a broader view, and to consider the cost of formal care and unpaid care, as well as other household costs.

To our knowledge, evidence of the total economic impact of SMA in Europe is scarce [11,13]. Thus, the main aim of this study is to fill two gaps in the information about SMA. First, by estimating the costs related to SMA from a societal perspective in the three European countries with the largest populations: Germany, France, and the UK. Secondly, by studying the HRQOL of SMA patients and their caregivers.

## 2. Methods

### 2.1. Study Population

This was a cross-sectional study of patients diagnosed with SMA who received outpatient care at the time of the study in three different European countries: France, Germany, and the United Kingdom. The Strengthening the Reporting of Observational Studies in Epidemiology (STROBE) guidelines were followed in the study [14].

Data were collected from July 2015 to November 2015. Children/adolescents diagnosed with SMA were eligible. Thus, 86 children and their caregivers were included in the study. The caregivers completed the administered questionnaires to supply information about the use of public health and non-health resources. More detailed information about the design and procedure of the study is available elsewhere [11]. We defined informal caregiving as a heterogeneous personal service, composed of various specific tasks provided to cover the basic or instrumental needs of a person with limited autonomy. More precisely, it is a non-professional activity, in the sense that the people who provide this care do not enjoy recognized employment rights, including weekly schedules and rest periods. Likewise, although they may receive family assistance or a public subsidy, it is not usually a paid activity. Particularly, we considered informal caregivers to be those who reported that they provide at least one hour of care per day.

The survey was totally anonymous, a patient organization/registry contacted the patients, and their responses were not associated with any identifying data (name, ID, address, e-mail). However, the study was submitted to an ethical committee in Germany to have a justification that the project and the collection of the data was made according to the ethic conditions.

### 2.2. Health Outcomes for Patients and Caregivers

The health-related quality of life of both patients and their caregivers was analysed. For this purpose, we used the proxy version of the EuroQol 5-dimentions and 3-levels (EQ-5D-3L) for patients and EuroQol 5-dimentions and 5-levels (EQ-5D-5L) for caregivers. The main reason for using different versions of questionnaires was that the EQ-5D-5L is only validated for adult responders. These questionnaires are generic instruments that are used for assessing the quality of life by considering five different dimensions: mobility, self-care, everyday activities, pain/discomfort, and anxiety/depression [15]. The standardized reference values are 0 (death or equivalent to death) and 1 (perfect health), although negative values (health status worse than death) also are possible. In addition, the EQ-5D instruments include a Visual Analogue Scale (VAS) that represents values from 0 (the worst health status) to 100 (the best) when asking participants to rate their overall health on the day of the interview.

Two other instruments were used to analyse the degree of patients’ dependence and the burden of care for caregivers. First, the Barthel Index, which measures the (dis)ability to perform the activities of daily living, assessing the degree of dependence [16,17,18], going from 0 points (totally dependent) to 100 points (totally independent). Secondly, the Zarit burden interview (22-item version) was used to measure the burden borne by caregivers due to the tasks provided. In this case, caregivers are supposed to answer questions about how they regard the care. The total score ranges from 0 to 88, with scores under 21 corresponding to little or no burden and scores above 61 to severe burden [19].

### 2.3. Treatment of Costs

We used a prevalence approach from a societal perspective. In order to estimate the economic burden of SMA, several questionnaires were used to collect information about the utilisation of health and non-health resources in the six-month period prior to the study (except for hospital admissions, for which the period used was the 12 months prior to the study). Costs were extrapolated to show annual costs. Information about hospital admissions, emergencies, medical tests, drug consumption, visits to general practitioners (GPs) and specialists, medical material and healthcare transport was collected in each country, and national reference prices were applied. More precisely, for the UK, the National Tariff Payment System was used to evaluate performance by the English National Health Service. For Germany, the information was obtained from Rosenfluh Publikationen AG, which provides official information about the German health system. In the case of France, the information was obtained from the websites “la sécurité sociale française” and “eureka Santé par vidal”, which provide official information about the health system in France.

Regarding non-healthcare costs, information was obtained about the social services used as well as about the informal (non-professional) care. Thus, the number of caregiving hours was obtained from the questions about the time spent providing children with help to carry out the Activities of Daily Living (ADL). The number of caregiving hours was then assessed by the proxy good method [20,21,22]. This technique assesses the value of the care provided by considering how much it would cost if informal caregivers had to be replaced in the employment market by a close substitute. Professional care wages per hour were €23.88 in the UK, €12.02 in France and €17.40 in Germany, according to the public rates provided by each country. Finally, for social services utilization, the information was collected from the questionnaires about the use of programmed home care, day centres, supportive social work, occupational centres, respiratory physiotherapy, physiotherapy, occupational physiotherapy, information/advice/assessment, psychosocial care for families, residential centres, hydrotherapy, and respite in temporary stays. The unitary cost for each service was obtained from local official sources. All prices are based on the year 2014.

## 3. Results

A total of 34, 27, and 25 children with SMA, and their caregivers, completed the questionnaires in the United Kingdom, France and Germany respectively (Table 1). Most of the children with SMA were classified as type II (58% and 48% for the UK, France, and Germany respectively), and had average ages of 5.5, 6.1 and 9.5 respectively for the three countries considered. The majority of children went to ordinary schools, although there were quite a few of them who went to nursery schools, especially in Germany (20%).

We identified 56 caregivers, 75% of whom reported a positive number of caregiving hours. Most of them were females, mainly in France (94% vs. 78% and 64% in Germany and the UK, respectively) (Table 1). They were also, on average, older in Germany (42 years old compared to 41 and 36 years in the UK and France respectively). In relation to the daily number of caregiving hours, it was observed that in the UK the intensity was higher in comparison with France and Germany. Informal caregivers from the UK provided an average of 12.50 h per day, while in France and Germany people cared for 10.65 and 9.31 h per day respectively. Finally, even though in France the intensity of caregiving was lower (in comparison with the UK and Germany), the burden borne due to the care was higher. More precisely, French caregivers produced a Zarit score of 40.37 versus 26.63 and 21.33 in the UK and Germany, respectively.

Table 2 shows the health-related quality of life for both patients and their caregivers. Regarding children, it was observed that French children had a lower quality of life, with a utility score of 0.12, a figure quite similar to that of children in the UK, with 0.17. By contrast, German children had a significantly better quality of life with a time trade-off (TTO) of 0.53. The dimension with the worst results was that of self-care, in which 41.18% of children in the UK said that they were unable to wash or dress themselves (44.44% in the case of France and 32% in Germany). These results correspond with those obtained in caregivers, as the French had an average utility score of 0.39 (the lowest), compared with TTO scores of 0.85 and 0.80 in the UK and Germany respectively.

The annual average cost associated with SMA reached €54,295 in the UK, €32,042 in France, and €51,983 in Germany (Table 3). In the three countries analysed, direct non-healthcare costs ranged between 79–86% of the total cost associated with SMA. More precisely, the item with the highest weight among the total costs was the care provided by relatives (informal care). Nevertheless, although the weights of healthcare and non-healthcare costs as a proportion of the total cost associated with SMA were quite similar, the amounts of such costs differ among the countries considered in this study. What is more, the percentage healthcare and non-healthcare costs above the total cost in each country are quite similar while the absolute figures differ. For instance, main informal caregivers cost entails around 61–63% of the non-healthcare cost of each country, but their absolute figures differ significantly, ranging from 17,500€ (in France) to 27,500€ (in Germany). (Figure 1).

In the United Kingdom, the total cost was the highest one, at €54,295, and the direct non-healthcare costs were also the highest, at €43,214 (that is, 79% of the total cost) and with direct healthcare costs amounting to €11,081 (20.4%). Within direct healthcare costs, the cost of specialist visits was €4569 (8.4% of the total cost), of health material, €1958 (about 3.6% of the total cost), of hospitalization, €2219 (4.1% of the total cost), of medical tests, €874, of healthcare transport, €58€ and of GPs and Emergencies, €842. Finally, drugs were valued at €560. Regarding direct non-healthcare costs, the cost of informal care was estimated at €40,526 (74.6% of the total cost) per year, while the cost of social services was €2187 (almost 4% of the total cost).

France had the lowest total cost associated with SMA (€32,042). €4672 (14.6%) corresponded to direct healthcare costs, while direct non-healthcare costs amounted to €27,370 (representing 85.4%). Within the cost of direct healthcare, the cost of medical visits was the highest, at €1870 (5.8% of the total cost), followed by hospitalization, the cost of which reached €1229 (3.8%). Regarding the cost of direct non-healthcare—that of informal care was estimated at €25,619 (80.0% of the total cost) per year, and the cost of social service was €1029 (3.2%).

Germany and the UK had the highest estimated costs (€51,983 and €54,295 respectively). 86% of the total cost corresponded to direct non-healthcare costs (€44,670), while direct healthcare represented 14% (€7313). Within the cost of direct healthcare, that of hospitalization was the most relevant, at €3170€ (6%). Regarding direct non-healthcare cost, the care provided by main caregivers was valued at €27,436, and the estimated value of the care provided by other carers was €12,490. The cost of social services was €4380 (8% of the total cost) (Table 3).

In brief, the results showed that the economic impact of SMA involves 1.40 times the GDP per capita in Germany, 1.02 times the GDP per capita in France and 1.70 times the GDP per capita in UK. Then, the figures display that the weight of the economic impact that disease has is quite similar across the countries included.

A table with costs in PPP is included in the Appendix A to a better comparison of these countries.

## 4. Discussion

This study represents the first complete and realistic costing study to date of the burden of SMA patients in Europe. Particularly, our results show that SMA is a disease that has a great economic impact from the perspective of society. In the countries considered, the total costs range from €32,000 to €54,000 per person per year, depending on the country. These figures include both high expenditure on health (between €4,700 and €11,000 on average per person, depending on the country) and even higher non-health costs (between €27,000 and €45,000 per person, depending on the country). Even though spending on health is very relevant, the importance of the resources invested in social services must also be highlighted (Germany stands out at €4,400 per person per year). However, the cost of informal care stands out as the main cost item, oscillating between 75% and 80% of the total economic impact. In addition to the impact on health, therefore, most of the economic impact falls on families in the form of time spent on care.

Broadly, our estimates do not greatly differ from those made previously in Germany [13] and in Spain [12], but some figures need to be clarified. For instance, the total costs of SMA were €54,721 for German patients and €33,721 for Spanish patients (slightly different from ours). The main differences may be due to differences in the method of accounting for direct healthcare resources and in the economic assessment of informal care. First, the previous study carried out in Germany included, in the direct healthcare costs, the more expensive healthcare resources (such as artificial nutrition systems, rehabilitation services and respiratory management) that we did not include in either this study or the one carried out in Spain. Secondly, the method used to assess the cost of informal caregiving time was not the same as the method used in the previous German study. Klug et al. only included the economic assessment of informal care costs for non-working parents in order to avoid double accounting because they also estimated the loss of productivity of working parents (which we did not) [13].

Regarding the HRQOL, this study is the first that uses the EQ-5D instrument to estimate the utility index score associated with the state of health of SMA patients and their caregivers. The previous study carried out by Klug et al. used a different tool for estimates connected with this condition, and the results cannot be compared. However, the instrument used in our paper was the same as the one used previously in the Spanish study. In this sense, the utility score of Spanish SMA patients was a lot lower (0.16 vs. 0.53) than for German SMA patients. However, the VAS score results included in the EQ-5D did not show such huge differences (54 vs. 69). Meanwhile, the HRQOL of informal caregivers was also different in the utility index score (0.49 vs. 0.81) but very similar in the VAS results (69 vs. 71).

Another point that should be highlighted is the large difference, identified in our study, between the HRQOL of the patients in France and the UK compared with that of patients in Germany. It seems that neither the age of the patients nor the degree of disease progression explains the differences observed. Therefore, we can only point out this fact and leave open, as a line of future research, the analysis of HRQOL in several countries, with as many samples as possible and with questions specially designed to give understanding of this variability. The same can be said about the important differences identified between the HRQOL of French caregivers and that of their counterparts in the UK and Germany.

Among the main limitations of the analysis, we can mention the limited sample size. Obviously, since it is a disease of low prevalence, it is expected that large sample sizes will not be available. However, it must be recognized that, given the distribution of SMA types over the total samples, it was not possible to perform an additional analysis for each phase of disease progression. Thus, another possible limitation could be the fact that the collection of information on patients was though patients’ organizations, and this might entail a selection bias. However, this method for collecting data has been also performed in other studies focused on rare diseases [23,24,25,26]. Likewise, the design of our study was cross-sectional, and the questions related to the health and non-health resources were retrospective. Ideally, the study would be a prospective one, using a longitudinal cohort of people with SMA, thus avoiding, for instance, the recall bias. However, we used an ad-hoc questionnaire aimed at avoiding recall bias, and in which the participants could answer the questions with no time limit. On the other hand, we should take into account the fact that, in studies where the subjects are children, it is difficult to identify how much time of care was provided due the illness and how much time was provided due to the development of the child. In this sense, we included questions aimed at estimating the informal care time, with a specific statement, which highlighted that the time provided should be related only to the illness.

Among rare diseases, SMA is one of those that receive attention from the health authorities and from society as a whole. First, because of its social consequences worldwide, and secondly, because of its incidence, prevalence and consequences in terms of loss of quality of life, mortality and morbidity. In fact, the estimated cost of SMA proves to be higher than the social costs (i.e., informal care costs) of other rare diseases such as ataxia (€18,776, base year 2004) [23] and similar to those of the fragile X syndrome (€31,008, base year 2012) [24], amyotrophic lateral sclerosis (€36,194) [25], and Duchenne muscular dystrophy (€36,970, base year 2012) [26].

In brief, the economic assessment of the informal caregiving time provided due to SMA disease reached figures higher than 70% of the total costs. This means that the majority of resources needed by SMA patients seem to come from outside the healthcare system. Even though some authors have assessed how the activities performed by informal caregivers affect their wellbeing [27,28,29] or their promotion in the workplace [30], there is still a lack of evidence in the field of SMA. Consequently, further research, focusing on identifying the effects of care on the health, employment status and socio-familiar dimension of informal caregivers, is needed. This information would help decision-makers to understand the vast effect of this disease in society, beyond its consequences for patients and the healthcare system. For instance, the inclusion of public health strategies focused on respite services may mitigate the aftermath of this illness beyond the patients [31,32], even in other caregiving populations [33]. Furthermore, financial aid for low-income households would also help the families that have—in addition to the health-related problems, or work-related problems—suffered by the main informal caregiver, financial problems due to the expensive medical material that they require. In fact, the cost of the families’ medical material and out-of-pocket expenses (such as those for house adaptation) may increase the cost of this illness beyond the healthcare resources needed [34]. Therapeutic agents for the treatment of SMA have therefore been approved for clinical use or are now in ongoing clinical trials. An antisense oligonucleotide that affects splicing of the pre-mRNA from the SMN2 gene (nusinersen-Spinraza^®^) was approved by the U.S. Food and Drug Administration (FDA) in December 2016 and by the European Medicines Agency (EMA) in June 2017 [35]. A self-complementary adeno-associated virus serotype 9 (AAV9) SMN1 gene therapy (Onasemnogene Abeparvovec, ZolgenSMA^®^) was approved recently in May 2019. New disease trajectories and evolving phenotypes are observed with these treatments [36]. The high cost of these treatments raises concerns about access and equity that should also be considered in the total burden of the disease and in the evaluation of these treatments.

## 5. Conclusions

SMA produces considerable societal costs in France, Germany, and the UK due to its relevant economic impact and the deterioration in the HRQOL, not only of the patients, but also of their caregivers. For this reason, when designing and evaluating any strategy or intervention for this population, the economic impact should be considered, as well as the economic evaluation of new treatments in this field.

## Figures and Tables

**Figure 1 ijerph-17-05640-f001:**
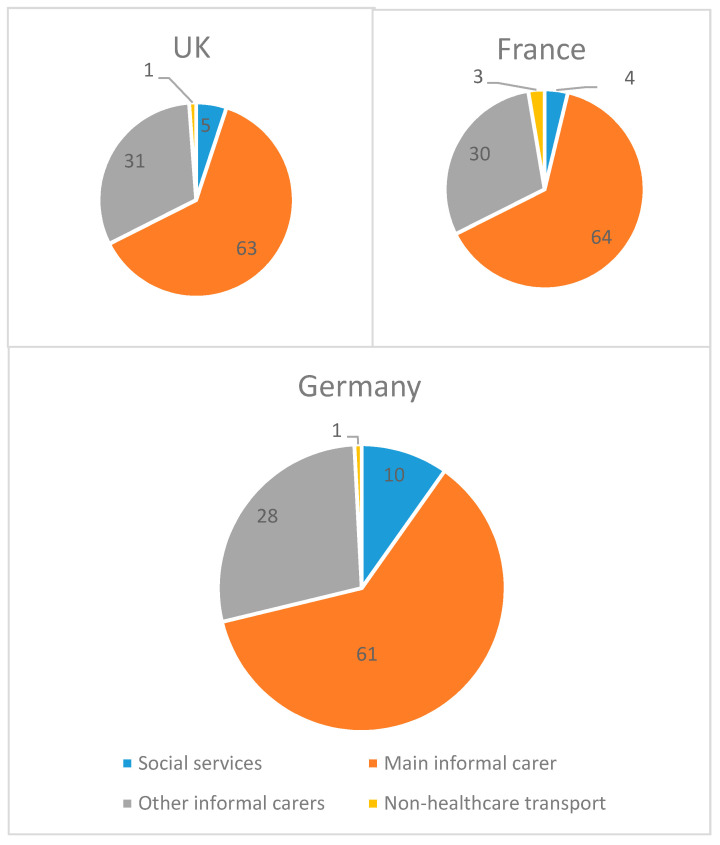
Weight of non-health care resources in the total costs by country. Source: Own preparation. Note: figures are percentages over the total costs.

**Table 1 ijerph-17-05640-t001:** Demographic characteristics of participants and their caregivers by country.

Characteristics	UK	France	Germany
	*N* (%)	*N* (%)	*N* (%)
Patients	*n* = 34	*n* = 27	*n* = 25
Type I	7 (20.59)	5 (18.52)	11 (44.00)
Type II	20 (58.82)	13 (48.15)	12 (48.00)
Type III	7 (20.59)	9 (33.33)	2 (8.00)
Gender (female)	17 (50.00)	16 (59.26)	18 (72.00)
Age (mean, SD)	5.55 (4.79)	6.19 (6.13)	9.52 (6.19)
**Education**			
Educated at an ordinary school	10 (29.41)	5 (18.52)	12 (48.00)
Educated at an ordinary centre with special sessions	6 (17.65)	8 (29.63)	4 (16.00)
Educated at a special education centre	2 (5.88)	1 (3.70)	4 (16.00)
Home-schooled	1 (2.94)	1 (3.70)	0
Nursery school	5 (14.71)	5 (18.52)	5 (20.00)
No education	7 (20.59)	4 (14.81)	0
NA	3 (8.82)	3 (11.11)	0
Barthel index (mean, SD)	35.93 (28.29)	34.33 (30.11)	47.66 (22.90)
Caregivers	*n* = 11	*n* = 16	*n* = 14
Gender (female)	7 (63.64)	15 (93.75)	11 (78.57)
Age (mean, SD)	41.09 (11.56)	36.13 (9.15)	42.53 (10.57)
Caregiving time (daily hours) (mean, SD) ^a^	12.50 (5.96)	9.31 (8.44)	10.65 (5.45)
Zarit caregiver’s burden, (mean, SD)	26.63 (13.39)	40.37 (16.10)	21.33 (18.33)
Risk of burnout (Zarit scale) (*n*, %) ^b^	0	1 (12.50)	0

Source: own preparation. ^a^ Number of daily hours allocated to informal caregiving was higher than 0. ^b^ People with a Zarit score equal to or higher than 55 points. SD: standard deviations. NA: no answer.

**Table 2 ijerph-17-05640-t002:** Health-related quality of life (HRQOL) of patients and caregivers by country.

	UK	France	Germany
	*n* = 34	*n* = 27	*n* = 25
	N (%)	N (%)	N (%)
Patients			
HRQOL (TTO social tariff score), mean (SD)	0.167 (0.277)	0.116 (0.285)	0.532 (0.335)
To be confined to bed	11 (32.35)	5 (18.52)	5 (20.00))
Unable to wash or dress himself/herself	14 (41.18)	12 (44.44)	8 (32.00)
Unable to perform usual activities	8 (23.53)	7 (25.93)	3 (12.00)
Extreme pain/discomfort	1 (2.94)	1 (3.70)	0
Suffer anxiety/mid depression	0	2 (7.41)	0
HRQOL (VAS score), mean (SD)	75.44 (19.36)	59.15 (29.84)	69.76 (13.42)
**Caregivers**	**UK** ***n* = 11**	**France** ***n* = 16**	**Germany** ***n* = 14**
HRQOL (TTO social tariff score), mean (SD)	0.852 (0.155) ^a^	0.396 (0.468) ^a^	0.800 (0.298) ^a^
Unable to walk	0	0	2 (15.38)
Unable to wash or dress himself/herself	0	4 (25.00)	1 (7.69)
Unable to perform usual activities	0	4 (25.00)	0
Extreme pain/discomfort	0	1 (6.25)	0
Suffer anxiety/mid depression	0	2 (12.50)	0
HRQOL (VAS score), mean (SD)	80.36 (17.01)	62.12 (33.41)	71.92 (14.20)

^a^ (SD). Source: own preparation. SD: standard deviations. TTO: Time trade-off. VAS: visual analogue scale.

**Table 3 ijerph-17-05640-t003:** Average annual costs by country (€) (2014).

	UK		France		Germany	
	*N* = 34		*N* = 27		*N* = 25	
	Mean (SD)	%	Mean (SD)	%	Mean (SD)	%
Drugs	560 (2085)	1.03%	14 (42)	0.04%	35 (106)	0.07%
Medical tests	874 (1223)	1.61%	384 (1375)	1.20%	158 (252)	0.30%
Medical visits	4569 (9908)	8.42%	1870 (2463)	5.84%	1954 (3656)	3.76%
Hospitalizations	2219 (5490)	4.09%	1229 (5260)	3.84%	3170 (4769)	6.10%
GP and Emergency	842 (3130)	1.55%	144 (274)	0.45%	617 (1966)	1.19%
Medical material	1958 (2428)	3.61%	990 (1770)	3.09%	1379 (1648)	2.65%
Healthcare transport	58 (270)	0.11%	41 (152)	0.13%	0	0.00%
**Direct healthcare costs**	**11,081 (10,764)**	**20.41%**	**4672 (7219)**	**14.58%**	**7313 (8636)**	**14.07%**
Social services	2187 (6197)	4.03%	1029 (2312)	3.21%	4380 (8044)	8.43%
***Direct non-healthcare formal costs***	***2187 (6197)***	***4.03%***	***1029 (2312)***	***3.21%***	***4380 (8044)***	**8.43%**
Main informal carer	27,012 (43,826)	49.75%	17,468 (22,550)	54.52%	27,436 (30,060)	52.78%
Other informal carers	13,516 (25,151)	24.89%	8151 (16,583)	25.44%	12,490 (17,050)	24.03%
***Direct non-healthcare informal costs***	***40,526 (60,016)***	***74.64%***	***25,619 (35,263)***	***79.95%***	***39,926 (42,047)***	**76.81%**
Non-healthcare transport ^a^	501 (1053)	0.92%	722 (1733)	2.25%	364 (381)	0.70%
**Direct non-healthcare costs**	**43,214 (61,139)**	**79.59%**	**27,370 (35,383)**	**85.42%**	**44,670 (45,063)**	**85.94%**
**TOTAL COST**	**54,295 (68,431)**	**100.00%**	**32,042 (38,303)**	**100.00%**	**51,983 (47,662)**	**100.00%**

^a^ This includes costs associated with non-healthcare transport and housing and vehicle adaptation. Source: own preparation SD: standard deviations. GP: general practitioner. Note: Bold/italic/shade data are the main results of each category

## Data Availability

The datasets generated and/or analysed during the current study are not publicly available due to [it is own by third party] but are available from the corresponding author on reasonable request.

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
