# Peer review of "The Economic Impact and Health-Related Quality of Life of Spinal Muscular Atrophy. An Analysis across Europe"

_ijerph, 2020, doi:10.3390/ijerph17165640_

Round 1
Reviewer 1 Report
The authors describe a study to assess the economic impact and health related quality of life of patients with spinal muscular atrophy in France, the UK and Germany. They include an assessment of both the direct healthcare costs, drugs, medical visits etc and direct non-healthcare costs incurred by informal unpaid carers.
They conclude that SMA results in considerable societal costs in the countries studied and that this should be considered when assessing strategies for intervention – particularly new treatments.
This is a topical issue where little evidence exists to guide policy makers faced with the evaluation of new genetic treatments such as nusinersen-Spinraza.
While the concept of the paper is to be welcomed, it has some very serious shortcomings:
- The use of English is very poor. I noted 12 occasions from line 54 to line 73 on p2 alone when it fell well below the standard required for publication in an international Journal. It represents the worst example of the use of English in any paper which I have reviewed and this certainly needs careful correction.
- The authors report on 86 children from the three countries mentioned without describing the number of patients with SMA in these countries. The authors suggest a prevalence of SMA of approximately 1:7,500 indicating that around 300 children are diagnosed each year in the three countries described. It is likely, as the responses to questionnaires were gathered anonymously via patient organisations, that there was significant selection bias from within a much larger patient group.
- The authors state on p3, line 102 that no ethical approval was need on the basis that the questionnaires were completed anonymously. While formal ethical committee approval may not have been needed, this would be for an ethics committee to determine and not the research group. In general when working in sensitive areas such as childhood disability an ethical opinion should always be sought before a study requiring parental participation is conducted – this is a serious flaw and would prevent publication in most journals.
- The authors indicate that a major element of the cost was based upon the direct non-healthcare costs that would have been paid to informal caregivers if they had been paid. These costs are calculated from a comparison in each country of the relevant professional wages paid for these activities. They indicate from these figures that the value of this work is twofold higher in the UK (24 euro/hour) when compared with France (12.00 euro/hour), p4 – line 143. This then is used to calculate the economic burden faced by parents in each country – Table 3. It seems very unlikely that this properly represents the situation in each country as it implies that parents caring for a child in the UK experience twice the costs per hour of parents in France.
Taken together, the lack of proper ethical assessment, the potentially unrepresentative selection of respondents, the doubts in relation to the costing assumptions of the value of the work and the very poor use of English, would not suggest that this paper could be accepted for publication.
Author Response
Authors have responded point by point to the reviewer`s comments in the attached file.

Reviewer 2 Report
Authors conducted an interesting study to investigate the economic impact and health related quality of life of patients with SMA patients across Europe. Their study revealed that SMA is a disease with a great economic impact from the perspective of society.
However, following points should be considered:
1) Lines 175 and 176: Authors mentioned that "The dimension with worse results was those related 175 to self-care, in which 77% of children in the UK said that there were unable to wash or dress 176 himself/herself (63% in the case of France and 44% in Germany)". However, in table 2, the percentages are different and are needed to be corrected in the text.
2) Lines 189-191: following sentence should be clarified:
Nevertheless, although the weight of healthcare and non-healthcare costs above the total cost associated with SMA was quite similar, the amount of such costs differs among countries considered in this study (figure 1).
Author Response

(The authors gave the same response as above.)

Reviewer 3 Report
General comments:
- Figure 1: Commas in charts, no unit. The figure should be signed under, not above.
- line 234- no reference number
- improve the quality of the figures
line 113 - in what categories do you understand the phrase "similar to death" - maybe it is worth exploring this issue?
line 114- similarly - worse than death? provide more details to understand the term
Table 3- expand the thought in parentheses
did the author check whether the value of money - especially purchasing power in the analyzed countries - is similar? Economic information about the economic situation could be useful - especially about the monetary situation to justify the comparison of these countries with each other
- I noticed incorrect punctuation or incorrectly formed sentences in several places, it would be advisable to review the text by a native speaker
Author Response

(The authors gave the same response as above.)
